# Tobacco smoking initiation among students in Samoa and health concerns

**Baljeet Singh**[1]◉*, **Shamal Shivneel Chand**[1]◉, **Hong Chen**[2]◉

**1** School of Accounting, Finance and Economics, The University of the South Pacific, Suva, Fiji, **2** School of Internet Economics and Business, Fujian University of Technology, Fujian, China

◉ These authors contributed equally to this work.
* singh_bl@usp.ac.fj

**Data Availability Statement:** The data used in this study are third party data from the World Health Organisation (https://extranet.who.int/ncdsmicrodata/index.php/catalog/748) and can be accessed following the protocol outlined in the Methods section.

## Abstract

### Background

High prevalence of tobacco smoking among young students remains a serious health concern given the positive association between smoking and NCDs. More recently, some studies also noted young smokers were more likely to get infected with COVID-19 compared to non-smokers. This study aims to assess the factors that influence smoking uptake among young students in Samoa. Findings from this study will provide valuable insight to policy-makers and health authorities on policies and strategies to combat smoking among youth in Samoa and the Pacific Island Countries (PICs).

### Methods

The 2017 Global Youth Tobacco Survey data of Samoa, available from the World Health Organization is used in the analysis. We use the multinominal logistic model to investigate the effects of socio-economic and demographics factors on young students' uptake of smoking in Samoa.

### Result

The main findings of this study indicate that sex, age, friendship, parental smoking, family discussion, outside influence, pocket money, and mother's education are important determinants of tobacco smoking initiation among youths in Samoa.

### Conclusion

Our findings contribute towards the evidence of the imperative health impact of friends, parents, and public smoking on students in Samoa. This warrants strategies that are effective in discouraging parents from smoking and implement measures that prevent smoking in public places. Moreover, educational efforts, particularly those that encourage more discussion at home settings on the harmful effects of smoking are strongly recommended. Parents are strongly encouraged to regularly monitor children's spending behaviour.

**Funding:** The authors received no specific funding for this work.

**Competing interests:** The authors have declared that no competing interests exists.

## Introduction

Tobacco smoking is injurious to health and one of the leading contributors of non-communicable diseases globally [1,2] including lung cancer [3], cardiovascular disease [4] and diabetes [5]. However, despite the known deleterious health effects of tobacco smoking being consistently flagged by the health authorities, the tobacco industry has continued to thrive, with the establishment of large-scale tobacco farms and industries [6]. Notably, majority of the global burden of smoking is in developing countries [7]. In the Pacific Island Countries (PICs), tobacco smoking is widespread across the population and is one of the leading cause of non-communicable diseases [8]. Despite large-scale consumption of tobacco, particularly among the youths [9] on one hand and rising incidents of NCDs such as cancer and heart disease on the other [10], factors that contribute to smoking in the young adult population in the region is less understood.

This study examines the behavioural factors that contribute to the adaptation of cigarette smoking by school students in the South Pacific Islands of Samoa. In Samoa, each year more than 165 people die due to tobacco-related diseases and more than 27,700 adults continue to smoke [11,12]. The prevalence of widespread tobacco smoking among the students in Samoa, make Samoa an ideal case study in the PICs. Moreover, this research is timely and vital as in a study, Prats-Uribe, Xie [13] noted that the likelihood of young smokers to get infected with COVID-19 were twice compared to non-smokers in UK, similarly, Adams, Park [14] found that young smokers were twice likely get infected with serious COVID-19 illness compared to non-smokers in US. Prats-Uribe, Xie [13] found no difference in mortality rate between young smokers and non-smokers was noted. There is lack of studies on long-term illness and impact of smoking on recoveries of illness from COVID-19, nevertheless, the higher risk of young smokers of been medically vulnerable, the medical policy intervention to address smoking among young smokers is likely to decrease their medical vulnerability to severe illnesses.

Numerous studies have examined the factors contributing to smoking adaptation behaviour globally. Andrea, Walter [15] in a study of Tuscan student nurses noted that a decrease in the availability of money, increase in the level of education, awareness of the harmful effects of tobacco smoking effectively contributed to a reduction in smoking among student nurses in Tuscany. Similarly, Biraghi and Tortorano [16] in a study of nursing students at the University of Milan noted that the smoking habits of parents, siblings and friends were positively associated with smoking among the nursing students.

Brenner and Scharrer [17] in a study of German medical students noted that maternal smoking was positively associated with smoking in students, while, paternal smoking behaviour and education background of the parents had no association with smoking in students. Dursun, Cesur [18] noted no association between increased levels of education and smoking behaviours in students in Turkey for both genders. In contrast, Kilic and Ozturk [19] noted a positive association between female education and smoking in Turkey since females became more independent with higher education. Ling, Neilands [20] in a study of the behaviour of young adult smoking behaviour in the United States of America noted that while education was negatively associated with smoking, however, exposure to smoking and tobacco smoking was positively associated with smoking among the young adults in the US. While the determinants of smoking behaviour are extensively researched, however, there is a lack of studies specific to small Pacific Island countries. Considering the conclusive evidence of factors that contribute to tobacco smoking, a country-specific study in the Pacific Island countries will provide insight on attributes that contribute to tobacco smoking in the region and will suggest policy measures that can be adopted by small island developing economies to discourage students from smoking.

This study examines the behavioural factors that contribute to the adoption of cigarette smoking in Samoa using the 2017 Global Youth Tobacco Survey data, available from the World Health Organization. We use the multinominal logistic model to investigate the effects of socio-economic and demographics factors on young students' uptake of smoking in Samoa. After accounting for missing data from the survey, a total of 1,754 observations were utilised for estimation purposes in Stata. Overall, the result of this study indicates that sex, age, friendship, parental smoking, family discussion, outside influence, pocket money, and mother's education are important determinants of tobacco smoking initiation among youths in Samoa.

This paper is structured as follows. Section 2 provides a brief description of the materials, data, and estimation procedures, followed by section 3 discussing the major findings and lastly, section 4 concludes with major findings and policy recommendations.

## Methods and materials

### Data

The study utilises data from Global Youth Tobacco Survey (GYTS) to investigate the leading and influential factors behind tobacco smoking adoption behaviour among youths in Samoa. The GYTS is a self-administered school-based survey designed to collect data and monitor tobacco use among the youths to devise and implement tobacco control and prevention programmes [21]. As part of the Global Tobacco Surveillance System, developed by the World Health Organization (WHO) and Centers for Disease Control and Prevention (CDC), GYTS is used as its data collection mechanism among youths. The latest GYTS survey in Samoa was carried out in 2017.

The survey had an overall response rate of 61.9 percent, with 2,076 students in grades 8–13 completing the survey [22]. Table 1 presents the descriptive summary of socio-economic and demographic variables for tobacco smoking among youths. The overall survey response was made up of 62.2 percent female students and 37.8 percent male students, of which 13 percent female students and 32.7 percent male students responded having tried smoking tobacco.

For the purpose of this study, smoking is defined as a student who tried or experimented with cigarette smoking, even one or two puffs. A score of 1 is given to student who tried or experimented with cigarette smoking, even one or two puffs, 0 otherwise.

Furthermore, students' attitudes to tobacco smoking are arguably influenced by a variety of factors such as demographic factors, peer group, income, and media [16,23,24]. In this study, we considered the probability of youth's initiating smoking status with the variables which were considered potential determinants of smoking initiation behaviour based on the existing literature.

Finally, we used multinomial logistic regression to measure and investigate factors influencing the attitude towards smoking among the students in the small Pacific Island Country of Samoa.

### Empirical model

In this study, we have used multinomial logit regression, expressed as follows:

$$logit(\pi) = \propto + \beta_1 X_1 + \beta_2 X_2 + \cdots + \beta_p X_p + e, \tag{1}$$

where $\pi$ is the probability of a student trying smoking, P(Y = 1) and $X_1$, $X_2$,. .., $X_p$ are the explanatory variables and $\beta_1$, $\beta_1$, . . .., $\beta_p$ are the logit coefficient of the explanatory variables. However, a more convenient form of estimating logit regression is the logarithm of odds which estimates the logistic transformation of the probability of success. Given that logit($\pi$) is

**Table 1. Summary statistics.**

| Indicators | Response | Tried smoking |
|---|---|---|
| *Sex* | | |
| Female | 1279 | 13.0% |
| Males | 778 | 32.7% |
| *Age* | | |
| 11 years or younger | 15 | 0.0% |
| 12 years old | 74 | 10.8% |
| 13 years old | 205 | 8.8% |
| 14 years old | 384 | 20.7% |
| 15 years old | 468 | 18.3% |
| 16 years old | 361 | 22.0% |
| 17 years old | 340 | 24.0% |
| 18 years or older | 223 | 31.9% |
| *During an average week, how much money do you have that you can spend on yourself, however you want?* | | |
| I usually do not have any spending money | 434 | 18.5% |
| Less than ST$30 | 1009 | 19.4% |
| ST$30.00–34.99 | 135 | 25.0% |
| ST$35.00–39.99 | 57 | 27.5% |
| ST$40.00–44.99 | 64 | 30.4% |
| ST$45.00–49.99 | 33 | 25.8% |
| ST$50 or more | 319 | 21.9% |
| *During the past 7 days, on how many days has anyone smoked in your presence, at any outdoor public place?* | | |
| 0 days | 788 | 12.6% |
| 1 to 2 days | 519 | 20.8% |
| 3 to 4 days | 259 | 28.6% |
| 5 to 6 days | 145 | 32.1% |
| 7 days | 343 | 25.6% |
| *During the past 30 days, did you see or hear any anti-tobacco media messages on television, radio, internet (e.g., Facebook, Google, WhatsApp, Twitter etc.), billboards, posters, newspapers, magazines, or movies?* | | |
| No | 443 | 18.3% |
| Yes | 1610 | 20.7% |
| *Has anyone in your family discussed the harmful effects of smoking tobacco with you?* | | |
| No | 648 | 21.8% |
| Yes | 1322 | 18.6% |
| *During school hours, how often do you see teachers smoking outdoors on school premises?* | | |
| Never | 423 | 14.7% |
| Sometimes | 747 | 19.3% |
| About every day | 662 | 26.0% |
| Don't know | 204 | 16.2% |
| *What level of education did your mother (stepmother or father's partner) complete?* | | |
| Primary education | 138 | 19.3% |
| Secondary education | 368 | 24.2% |
| Tertiary education | 701 | 19.8% |
| Vocational training | 137 | 24.6% |
| Don't know | 671 | 17.5% |

(*Continued*)

**Table 1.** (Continued)

| Indicators | Response | Tried smoking |
|---|---|---|
| ***Do your parents smoke tobacco?*** | | |
| None | 1228 | 17.0% |
| Both | 275 | 24.4% |
| Father only | 390 | 24.0% |
| Mother only | 112 | 25.9% |
| Don't know | 36 | 48.3% |
| ***Do any of your closest friends' smoke tobacco?*** | | |
| None of them | 1427 | 12.3% |
| Some of them | 408 | 35.9% |
| Most of them | 101 | 46.2% |
| All of them | 75 | 48.6% |

Source: GYTS and Authors' calculation. The descriptive statistics is derived from the full sample which is 2076 observation, however, some categories may not add upto 2076 due to the missing values in that particular categories. After accounting for all missing values in the sample, we have only used 1754 observation for the regression analysis.

ln(odds), which is expressed as $ln\frac{\pi}{1-\pi}$, the multinomial logit model is expressed as follow:

$$ln\left[\frac{\pi(x)}{1-\pi(x)}\right] = \propto + \beta_1 X_1 + \beta_1 X_1 + \cdots + \beta_p X_p + e \qquad (2)$$

Where $X_1, X_2, \ldots, X_p$ are the explanatory variables and $\beta_1, \beta_1, \ldots, \beta_p$ are the logit coefficient of the explanatory variables. By taking the exponential of the logit coefficient, we computed the odds ratio. The odds are the ratio is the probability of smoking to the probability of not smoking. An odds ratio is the ratio of two odds. When the value of the odds ratio is higher than 1, it implies that the odds of success or an event taking place increases for one unit increase in the explanatory variable. Similarly, when the value of the odds ratio is less than 1, it implies that the odds of success or an event taking place decreases for one unit increase in the explanatory variable A odds ratio equal to 1 implies that there is no association between odds of success and explanatory variable. The regression analysis was carried out using Stata software.

## Findings and discussion

### Results

The estimated result using the multinomial logistic regression is provided in Table 2. We noted that factors such as smoking among teachers in schools and media have no statistical impact on smoking behaviour among school students in Samoa.

Overall, spending capacity and money on hand to some extent had a significant impact on tobacco smoking. We noticed that only pocket money between ST$40.00 to ST$44.99 and ST$45.00 to ST$49.99 had a positive and significant impact on smoking. The odds of smoking for a student with ST$40.00 to ST$44.99 pocket money were 1.94 times the odds of smoking for a student with no pocket money. Similarly, the odds of smoking for a student with ST$ 45.00 to ST$49.99 pocket money were 2.22 times the odds of smoking for a student with no pocket money. Pocket money for categories below ST$45.00 did have not have any significant influence on smoking experimentation. This indicates that a certain threshold level of pocket

**Table 2. Regression results.**

| Independent Variables | Estimate | Odds Ratio | 95% CI |
|---|---|---|---|
| Constant | -4.06 | 0.02*** | (0.01,0.04) |
| *Sex* | | | |
| Female (reference) | | | |
| Male | 0.90 | 2.45*** | (1.87,3.22) |
| *Age* | 0.13 | 1.13*** | (1.04,1.24) |
| *During an average week, how much money do you have that you can spend on yourself, however you want?* | | | |
| I usually do not have any spending money (reference) | | | |
| Less than ST$30.00 | 0.22 | 1.24 | (0.88,1.76) |
| ST$30.00–34.99 | 0.30 | 1.35 | (0.76,2.42) |
| ST$35.00–39.99 | 0.13 | 1.14 | (0.49,2.63) |
| ST$40.00–44.99 | 0.67 | 1.95* | (0.93,4.06) |
| ST$45.00–49.99 | 0.80 | 2.22* | (0.88, 5.63) |
| ST$50.00 or more | 0.14 | 1.15 | (0.74,1.80) |
| *During the past 7 days, on how many days has anyone smoked in your presence, at any outdoor public place?* | | | |
| 0 days (reference) | | | |
| 1 to 2 days | 0.51 | 1.66*** | (1.16,2.37) |
| 3 to 4 days | 0.72 | 2.06*** | (1.36,3.14) |
| 5 to 6 days | 0.93 | 2.53*** | (1.55,4.12) |
| 7 days | 0.48 | 1.61** | (1.08,2.41) |
| *During the past 30 days, did you see or hear any anti-tobacco media messages on television, radio, internet (e.g., Facebook, Google, WhatsApp, Twitter etc.), billboards, posters, newspapers, magazines, or movies?* | | | |
| No (reference) | | | |
| Yes | 0.05 | 1.05 | (0.74,1.49) |
| *Has anyone in your family discussed the harmful effects of smoking tobacco with you?* | | | |
| No (reference) | | | |
| Yes | -0.27 | 0.77* | (0.58,1.02) |
| *Do any of your closest friends' smoke tobacco?* | | | |
| None of them (reference) | | | |
| Some of them | 1.10 | 3.01*** | (2.24,4.05) |
| Most of them | 0.97 | 2.63*** | (1.51,4.59) |
| All of them | 1.41 | 4.09*** | (2.26,7.40) |
| *During school hours, how often do you see teachers smoking outdoors on school premises?* | | | |
| Never (Reference) | | | |
| Sometimes | 0.11 | 1.12 | (0.76,1.65) |
| About every day | 0.32 | 1.37 | (0.93,2.04) |
| Do not Know | -0.08 | 0.92 | (0.52,1.63) |
| *What level of education did your mother (stepmother or father's partner) complete?* | | | |
| Primary education (reference) | | | |
| Secondary Education | 0.63 | 1.87* | (0.98,3.56) |
| Tertiary Education | 0.32 | 1.38 | (0.75,2.56) |
| Vocational Trainings | 0.65 | 1.91* | (0.90,4.09) |
| Do not Know | 0.28 | 1.32 | (0.71,2.47) |
| *Do your parents smoke tobacco?* | | | |
| None (reference) | | | |
| Both | 0.23 | 1.26 | (0.86,1.86) |
| Father | 0.41 | 1.52** | (1.09,2.12) |
| Mother | 0.29 | 1.34 | (0.76,2.35) |

*(Continued)*

**Table 2.** (Continued)

| Independent Variables | Estimate | Odds Ratio | 95% CI |
|---|---|---|---|
| Do not Know | 1.42 | 4.13*** | (1.53,11.13) |
| Pseudo R$^2$ = 0.1435 | | | |
| Hosmer-Lemeshow Chi2(8) = 10.27 Prob>0.25 | | | |

Note

***, ** and * represent 1%, 5% and 10% significance levels.

money is needed to motivate students to smoke. Similarly, we did not find any significant difference between the odds of smoking between a student with no money and a student with highest level of pocket money. This result is not surprising given that a student who does not have the pocket money do not have the means to buy smoke, while a student with too much pocket money can afford a better alternative form of pleasure to smoking. Students with too much pocket money can also reflect a better socio-economic background and hence are less likely to indulge in smoking.

Furthermore, the age of students is also statistically significant in explaining smoking behaviour among students. With a one-year increase in age, the odds of smoking among students increases by 1.13. This indicates that more students are likely to try out smoking as they get older. This result is within expectation and in line with the findings from other studies. This result is consistent with Chen and Millar [25] in the context of Canada who found that age plays a critical indicator for the adoption of smoking habits among adults and with increasing age, cigarette consumption becomes more prevalent.

The odds of a male student smoking in Samoa is 2.45 times larger compared to the odds of a female student smoking. This is as expected since the smoking prevalence among the male population in Samoa is significantly higher compared to females in all age groups [26].

Concerning peer influence, we observed that students are more likely to initiate smoking if their close friends also indulge in smoking. In the survey, students were asked to respond whether their friends smoked in the order: none of them, some of them, most of them and all of them, where for analysis purposes, none of them was chosen as the reference level. According to the results, the odds of a student trying to smoke when some of their friends smoked were 3.01 times larger as compared to when none of their friends smoked. Further, the odds of a student smoking if most of their friends or all their friends smoked were 2.63 and 4.09 times larger, respectively, than the odds of a student who had none of the friends smoking during the survey period. This result indicates that friends, especially close friends are one of the major influencers and peer pressure groups towards trying or experimenting with tobacco smoking, which eventually leads to addiction after some time. The findings further indicate that when most of the friends smoke, they are likely to pressure non-smokers to take up smoking. This result is similar to study findings from Greece [27], Ethiopia [28] and Bangladesh [29]. Pacific islanders have a culture of sharing cigarettes and with more friends smoking, it reduces the financial burden as the cost of tobacco and cigarettes are shared among a larger group of students.

With respect to the influence of the mother's education level on a student's smoking behaviour, we find that the odds of a student smoking are 1.87 times higher when the mother only has secondary education compared to a mother with primary education as the base level. The odds of a student smoking are 1.91 times higher when the mother has vocational training compared to a mother with primary education. Mother with tertiary education level qualification

side has insignificant effects on whether the student has tried or experimented with tobacco smoking. The result indicates that mothers with a secondary or vocational level qualification are more likely to engage in labour intensive and tedious occupation and therefore are less likely to spend quality time with their children, while mothers with primary education are more likely to stay home and have quality time with children. On the other hand, mothers with tertiary qualification have better education and occupation to be able to spend quality time with children and educate on the harmful effects of tobacco smoking.

With respect to the influence of parent's smoking behaviour on children's smoking, we noted that smoking among fathers has a significant impact on whether their child also starts smoking. The odds of a student smoking whose father smoked were 1.52 times larger than the odds of a student with no parents smoking. However, smoking among both parents and only among mothers is found to be insignificant at a 5 percent significance level. The results overall show that students see fathers as the role model and major influencer in their teenage and youth years and have a higher probability of adopting his habits such as smoking tobacco, similar to findings from Stanton, Papandonatos [30].

Furthermore, we noted that discussion within families on the harmful effects of smoking has a negative relationship with tobacco smoking adoption among students. We observed that the odds of a student smoking whose parents had earlier discussed the harmful effects of smoking within the family was only 0.77 times compared to the odds of a student experimenting with tobacco without any family discussion on the harmful effects of tobacco smoking. This implies that raising more awareness among the parents on the harmful effects of smoking and encouraging them to discuss with their children is likely to reduce the tobacco smoking adoption among students in Samoa.

We further performed Hosmer-Lemeshow Goodness-of-Fit Test (see Table 2) to find out if our model is an excellent fit. Following the standard practice, see Liu [31], a group (10) option was used for our purpose. In our model, Hosmer-Lemeshow Chi-square test has a value of 10.27 with the associated p-value of 0.25 which is insignificant and therefore confirms that the model fits the data well.

## Discussion

This study provides evidence that peer pressure, home and community environment are critical factors to explaining smoking initiation among students in Samoa. On one hand, we noted that students were more likely to smoke if their father or friends were smoking or who witness others smoking in public. On the other hand, we identified that students whose parents discussed the harmful effects of smoking at home were less likely to smoke. These two observations can plausibly provide a crucial avenue to address the smoking problem among students in Samoa. Recreational activities that promote a smoker-friendly environment such as bars and clubs, sporting events and game centres profit from this weakness. Direct policy intervention that bars smoking in public places such as sporting venues, parks, public transport, religious places, town, and cities can crucially play an important role in limiting smoking uptake in Samoa. Moreover, public health intervention that encourages more parents to discuss harmful effects of smoking with their children in Samoa will discourage students from taking up smoking.

The positive association between fathers' smoking behaviour and the absence of any association between maternal smoking paints a significant divergent role of fathers and mothers in child's education in Samoa. Student homes and social places are vital settings for public health intervention to reduce smoking uptake among the students in Samoa. Targeting parents, particularly fathers who smoke tobacco through various health intervention is likely to reduce the child's uptake of smoking.

It is equally important that more and more women should be encouraged to take up tertiary qualification. It is noted that students whose mother have secondary or vocational education are more likely to smoke. This could be directly related to the nature of tedious work that women do with this level of qualification. Encouraging more women to complete the tertiary qualification will reduce the number of mothers having secondary or vocational qualification and will therefore reduce student's uptake of smoking.

Our findings also confirm that the availability of pocket money is also an important determinant of smoking uptake among students. Through various campaigns and counter-marketing strategies, parents should be educated on potential abuses of pocket money and the need to monitor the spending habits of students. Limiting and strict monitoring of pocket money can reduce smoking uptake among the students in Samoa.

Reasonably, we observed that media does not have any impact on the uptake of smoking in Samoa. Samoa is a small low-income country and many households do not have adequate access to any form of decent media and internet penetration remains low among the population to access valuable information [32]. Therefore, as observed media is less likely to play any critical role in shaping the behaviours of the majority of Pacific Islanders.

Male students are more likely to adopt tobacco smoking as compared to female students, henceforth, more policy interventions should be targeted towards the male population group such as the implementation of educational programmes and support groups to disseminate information on the dangers of smoking.

## Conclusion

Tobacco smoking has become a major problem among the small island developing states, especially its high consumption among the young and vulnerable population groups. Tobacco smoking has been the major risk factor for many underlying health-related problems and non-communicable diseases. The findings of this study provide relevant policy recommendations to counter the high rate of tobacco smoking adoption among youths in Samoa. This study immensely contributes towards the literature on the major determinants of tobacco smoking among youths in the small island developing state of Samoa.

In summary, our findings contribute towards the evidence of the imperative health impact of friends, parents and public smoking on students in Samoa. This warrants strategies that are more effective to discourage parents from smoking and measure to prevent smoking in public places, which exposes the vulnerability of the students. Moreover, educational efforts, particularly those that encourage more discussion at home on the harmful effects of smoking and create more awareness on the need to monitor children's spending behaviour are strongly recommended.

Despite media being not a significant factor in explaining whether student tried smoking, the impact of media should not be neglected since it is one of the major drivers of smoking in other countries [33,34]. Samoa should use Government and privately-owned media platforms to create mass media smoking cessation campaigns, especially specific campaigns targeted towards youths. Mass media interventions are successful in effectively changing smoking behaviour among adults in other country contexts [35,36].

## Author Contributions

**Conceptualization:** Baljeet Singh, Shamal Shivneel Chand, Hong Chen.

**Data curation:** Baljeet Singh, Shamal Shivneel Chand, Hong Chen.

**Formal analysis:** Baljeet Singh, Shamal Shivneel Chand, Hong Chen.

**Investigation:** Baljeet Singh, Shamal Shivneel Chand, Hong Chen.

**Methodology:** Baljeet Singh, Shamal Shivneel Chand, Hong Chen.

**Project administration:** Baljeet Singh, Shamal Shivneel Chand, Hong Chen.

**Resources:** Baljeet Singh, Shamal Shivneel Chand, Hong Chen.

**Software:** Baljeet Singh, Shamal Shivneel Chand, Hong Chen.

**Supervision:** Baljeet Singh, Shamal Shivneel Chand, Hong Chen.

**Validation:** Baljeet Singh, Shamal Shivneel Chand, Hong Chen.

**Visualization:** Baljeet Singh, Shamal Shivneel Chand, Hong Chen.

**Writing – original draft:** Baljeet Singh, Shamal Shivneel Chand, Hong Chen.

**Writing – review & editing:** Baljeet Singh, Shamal Shivneel Chand, Hong Chen.

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
