## [Decision Letter · Decision Letter 0]

9 Aug 2021

PONE-D-21-22399

Tobacco Smoking Initiation among students in Samoa and Health Concerns

PLOS ONE

Dear Dr. Singh,

Thank you for submitting your manuscript to PLOS ONE. After careful consideration, we feel that it has merit but does not fully meet PLOS ONE’s publication criteria as it currently stands. Therefore, we invite you to submit a revised version of the manuscript that addresses the points raised during the review process.

All the comments raised by the reviwer should be adressed point to point appropriately.

We look forward to receiving your revised manuscript.

Kind regards,

Gausal A Khan, Ph.D;CSci, FRSB

Academic Editor

PLOS ONE

Journal Requirements:

Additional Editor Comments :

Please repley the comments point to point rasied by the reviwer

Reviewers' comments:

Reviewer's Responses to Questions

**Comments to the Author**

1. Is the manuscript technically sound, and do the data support the conclusions?

Reviewer #1: Partly

Reviewer #2: Yes

2. Has the statistical analysis been performed appropriately and rigorously? 

Reviewer #1: N/A

Reviewer #2: Yes

3. Have the authors made all data underlying the findings in their manuscript fully available?

Reviewer #1: Yes

Reviewer #2: Yes

4. Is the manuscript presented in an intelligible fashion and written in standard English?

Reviewer #1: Yes

Reviewer #2: Yes

5. Review Comments to the Author

Reviewer #1: This study is interesting and have a great impact on public health threat, especially for young students in Samoa and the Pacific Island Countries. The data also insight on the world economy and policy making suggestions.

Right now, the whole world is affected with COVID-19 infection and study reported that young smokers were more got infected with COVID-19 compared with non-smokers.

1. How does COVID-19 infection relate to the mortality rate with young smoker VS Non-smokers

2. How does the recovery from COVID-19 infection affected the young smoker?

Reviewer #2: In this manuscript “Tobacco Smoking Initiation among students in Samoa and Health Concerns“ Singh et al. demonstrated a living problem of socio-economic, education and culture in context of tobacco smoking in the young generation of Samoa.

The study has a significant impact apropos of socio-economic background, which is not only a problem of a particular area but it is the issue of many countries in all over the world.

Here, authors demonstrated how home and friend circle environment, parental smoking, pocket money and mother’s education have an impact on youths in smoking in Samoa. They used empirical formula for the analysis: multinomial logit regression.

However, I have few minor question:

1) Introduction is too long, authors can decrease the volume.

2) In table -1, sex, age should be bold

3) Authors used empirical model in the method section, they should mention the use of software/link.

4) In conclusion, in a few lines, authors may emphasize and write their main findings and followed by how to overcome that problem.

6. PLOS authors have the option to publish the peer review history of their article (what does this mean?). If published, this will include your full peer review and any attached files.

Reviewer #1: No

Reviewer #2: No

---

## [Author Response · Author response to Decision Letter 0]

5 Sep 2021

Response to Reviewers’ comments

We are very thankful to the reviewers for their detailed and insightful comments and suggestions. It has helped us to improve the manuscript significantly. 

Comments to the Author

1. Is the manuscript technically sound, and do the data support the conclusions?

Reviewer #1: Partly

This study is based on survey data, which was collected through scientific methodology developed by World Health Organization (WHO). This is a normal practice in this kind of studies. We have done some revisions to the analysis section to provide a more technical tone to the script. We have used data collected by WHO.

Reviewer #2: Yes

2. Has the statistical analysis been performed appropriately and rigorously? 

Reviewer #1: N/A

The statistical analysis has been conducted using the multinominal logistic regression which is robust across different scenarios. In addition to this, we have also performed the Hosmer-Lemeshow Goodness-of-Fit test to show the excellent fit of the regression model. The test overall shows that the model fits the data well from which the discussions are outlayed.

Reviewer #2: Yes

3. Have the authors made all data underlying the findings in their manuscript fully available?

Reviewer #1: Yes

Reviewer #2: Yes

4. Is the manuscript presented in an intelligible fashion and written in standard English?

Reviewer #1: Yes

Reviewer #2: Yes

5. Review Comments to the Author

Reviewer #1: This study is interesting and have a great impact on public health threat, especially for young students in Samoa and the Pacific Island Countries. The data also insight on the world economy and policy making suggestions.

Right now, the whole world is affected with COVID-19 infection and study reported that young smokers were more got infected with COVID-19 compared with non-smokers.

1. How does COVID-19 infection relate to the mortality rate with young smoker VS Non-smokers

We have cited and included some papers analyzing on the above question in this paper. 

Prats-Uribe, Xie (2021) find no difference in mortality rate between young smokers and non-smokers, however young smokers were twice likely to get infected with serious COVID-19 illness compared to non-smokers, an important finding from the United States.

2. How does the recovery from COVID-19 infection affected the young smoker?

To our knowledge, there is no studies on long-term illness and impact of smoking on recoveries of illness from COVID-19, nevertheless, the higher risk of young smokers of been medically vulnerable, the medical policy intervention to address smoking among young smokers is likely to decrease their medical vulnerability to severe illnesses. This part as been included in the paper.

Reviewer #2: In this manuscript “Tobacco Smoking Initiation among students in Samoa and Health Concerns“ Singh et al. demonstrated a living problem of socio-economic, education and culture in context of tobacco smoking in the young generation of Samoa.

The study has a significant impact apropos of socio-economic background, which is not only a problem of a particular area but it is the issue of many countries in all over the world.

Here, authors demonstrated how home and friend circle environment, parental smoking, pocket money and mother’s education have an impact on youths in smoking in Samoa. They used empirical formula for the analysis: multinomial logit regression.

However, I have few minor question:

1) Introduction is too long, authors can decrease the volume.

As recommended, we have reduced the size of the introduction section by removing few sentences and condensing main ideas.

2) In table -1, sex, age should be bold

As recommended, we have made the above changes in the paper.

3) Authors used empirical model in the method section, they should mention the use of software/link.

Authors have used Stata for conducting statistical analysis. This information has been included in the empirical model section.

4) In conclusion, in a few lines, authors may emphasize and write their main findings and followed by how to overcome that problem.

6. PLOS authors have the option to publish the peer review history of their article (what does this mean?). If published, this will include your full peer review and any attached files.

Do you want your identity to be public for this peer review? For information about this choice, including consent withdrawal, please see our Privacy Policy.

Reviewer #1: No

Reviewer #2: No

---

## [Decision Letter · Decision Letter 1]

4 Oct 2021

Tobacco Smoking Initiation among students in Samoa and Health Concerns

PONE-D-21-22399R1

Dear Dr. Singh,

We’re pleased to inform you that your manuscript has been judged scientifically suitable for publication and will be formally accepted for publication once it meets all outstanding technical requirements.

Kind regards,

Gausal A Khan, Ph.D;CSci, FRSB

Academic Editor

PLOS ONE

Additional Editor Comments (optional):

Reviewers' comments:

Reviewer's Responses to Questions

**Comments to the Author**

1. If the authors have adequately addressed your comments raised in a previous round of review and you feel that this manuscript is now acceptable for publication, you may indicate that here to bypass the “Comments to the Author” section, enter your conflict of interest statement in the “Confidential to Editor” section, and submit your "Accept" recommendation.

Reviewer #1: All comments have been addressed

Reviewer #2: (No Response)

2. Is the manuscript technically sound, and do the data support the conclusions?

Reviewer #1: Partly

Reviewer #2: Yes

3. Has the statistical analysis been performed appropriately and rigorously? 

Reviewer #1: N/A

Reviewer #2: Yes

4. Have the authors made all data underlying the findings in their manuscript fully available?

Reviewer #1: Yes

Reviewer #2: (No Response)

5. Is the manuscript presented in an intelligible fashion and written in standard English?

Reviewer #1: Yes

Reviewer #2: (No Response)

6. Review Comments to the Author

Reviewer #1: The manuscript elicited attention on behavior and pattern of public smoking among family members and friends have great impact on students in Samoa. The manuscript used the data from the Global Youth Tobacco Survey (GYTS) and also analysis was done using multinomial logistic regression to find out the various factors which influence the students in the small Pacific Island Country of Samoa towards smoking. The manuscript is well described all parts and also addressed all comments. I am highly recommend this manuscript for publication.

Reviewer #2: (No Response)

7. PLOS authors have the option to publish the peer review history of their article (what does this mean?). If published, this will include your full peer review and any attached files.

Reviewer #1: No

Reviewer #2: No

---

## [Editor Report · Acceptance letter]

21 Oct 2021

PONE-D-21-22399R1 

Tobacco Smoking Initiation among students in Samoa and Health Concerns 

Dear Dr. Singh:

I'm pleased to inform you that your manuscript has been deemed suitable for publication in PLOS ONE. Congratulations! Your manuscript is now with our production department. 

Kind regards, 

on behalf of

Dr. Gausal A Khan 

Academic Editor

PLOS ONE